# BETTY: AN AUTOMATIC DIFFERENTIATION LIBRARY FOR MULTILEVEL OPTIMIZATION

**Sang Keun Choe**[1]    **Willie Neiswanger**[2]    **Pengtao Xie**[3,4*]    **Eric Xing**[1,4*]

[1]Carnegie Mellon University    [2]Stanford University    [3]UCSD    [4]MBZUAI    [*]Equal contribution
{sangkeuc,epxing}@cs.cmu.edu, neiswanger@cs.stanford.edu, p1xie@ucsd.edu

## ABSTRACT

Gradient-based multilevel optimization (MLO) has gained attention as a framework for studying numerous problems, ranging from hyperparameter optimization and meta-learning to neural architecture search and reinforcement learning. However, gradients in MLO, which are obtained by composing best-response Jacobians via the chain rule, are notoriously difficult to implement and memory/compute intensive. We take an initial step towards closing this gap by introducing BETTY, a software library for large-scale MLO. At its core, we devise a novel dataflow graph for MLO, which allows us to (1) develop efficient automatic differentiation for MLO that reduces the computational complexity from $\mathcal{O}(d^3)$ to $\mathcal{O}(d^2)$, (2) incorporate systems support such as mixed-precision and data-parallel training for scalability, and (3) facilitate implementation of MLO programs of arbitrary complexity while allowing a modular interface for diverse algorithmic and systems design choices. We empirically demonstrate that BETTY can be used to implement an array of MLO programs, while also observing up to 11% increase in test accuracy, 14% decrease in GPU memory usage, and 20% decrease in training wall time over existing implementations on multiple benchmarks. We also showcase that BETTY enables scaling MLO to models with hundreds of millions of parameters. We open-source the code at https://github.com/leopard-ai/betty.

## 1  INTRODUCTION

Multilevel optimization (MLO) addresses nested optimization scenarios, where upper level optimization problems are constrained by lower level optimization problems following an underlying hierarchical dependency. MLO has gained considerable attention as a unified mathematical framework for studying diverse problems including meta-learning (Finn et al., 2017; Rajeswaran et al., 2019), hyperparameter optimization (Franceschi et al., 2017), neural architecture search (Liu et al., 2019), and reinforcement learning (Konda & Tsitsiklis, 1999; Rajeswaran et al., 2020). While a majority of existing work is built upon bilevel optimization, the simplest case of MLO, there have been recent efforts that go beyond this two-level hierarchy. For example, (Raghu et al., 2021) proposed trilevel optimization that combines hyperparameter optimization with two-level pretraining and finetuning. More generally, conducting joint optimization over machine learning pipelines consisting of multiple models and hyperparameter sets can be approached as deeper instances of MLO (Garg et al., 2022; Raghu et al., 2021; Somayajula et al., 2022; Such et al., 2020).

Following its increasing popularity, a multitude of optimization algorithms have been proposed to solve MLO. Among them, *gradient-based (or first-order)* approaches (Pearlmutter & Siskind, 2008; Lorraine et al., 2020; Raghu et al., 2021; Sato et al., 2021) have recently received the limelight from the machine learning community, due to their ability to carry out efficient high-dimensional optimization, under which all of the above listed applications fall. Nevertheless, research in gradient-based MLO has been largely impeded by two major bottlenecks. First, implementing gradients in multilevel optimization, which is achieved by composing best-response Jacobians via the chain rule, requires both programming and mathematical proficiency. Second, algorithms for best-response Jacobian calculation, such as iterative differentiation (ITD) or approximate implicit differentiation (AID) (Grazzi et al., 2020), are memory and compute intensive, as they require multiple forward/backward computations and oftentimes second-order gradient (*i.e.* Hessian) information.

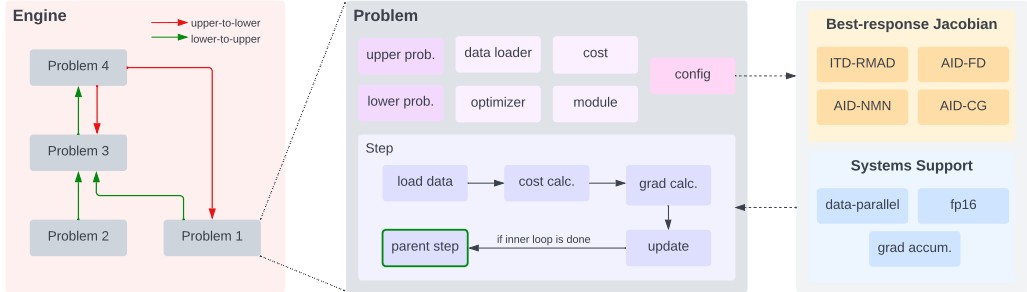

Figure 1: In `Engine` (left), users define their MLO program as a hierarchy/graph of optimization problems. In `Problem` (middle), users define an optimization problem with a data loader, cost function, module, and optimizer, while upper/lower level constraint problems (*i.e.* $\mathcal{U}_k$, $\mathcal{L}_k$) are injected by `Engine`. The "step" function in `Problem` serves as the base of gradient-based optimization, abstracting the one-step gradient descent update process. Finally, users can easily try out different best-response Jacobian algorithms & system features (right) via `Config` in a modular manner.

In recent years, there has been some work originating in the meta-learning community on developing software libraries that target some aspects of gradient-based MLO (Blondel et al., 2021; Deleu et al., 2019; Grefenstette et al., 2019). For example, *JAXopt* (Blondel et al., 2021) provides efficient and modular implementations of AID algorithms by letting the user define a function capturing the optimality conditions of the problem to be differentiated. However, *JAXopt* fails to combine the chain rule with AID to support general MLO programs beyond a two-level hierarchy. Similarly, *higher* (Grefenstette et al., 2019) provides several basic primitives (*e.g.* making PyTorch's (Paszke et al., 2019) native optimizers differentiable) for implementing ITD/AID algorithms, but users still need to manually implement complicated internal mechanisms of these algorithms as well as the chain rule to implement a given instance of MLO. Furthermore, most existing libraries do not have systems support, such as mixed-precision and data-parallel training, that could mitigate memory and computation bottlenecks. As a result, gradient-based MLO research built upon these libraries has been largely limited to simple bilevel optimization and small-scale setups.

In this paper, we attempt to bridge this gap between research and software systems by introducing BETTY, an easy-to-use and modular automatic differentiation library with various systems support for large-scale MLO. The main contributions of this paper are as follows:

1. We develop an efficient automatic differentiation technique for MLO based on a novel interpretation of MLO as a special type of dataflow graph (Section 3). In detail, gradient calculation for each optimization problem is automatically carried out by iteratively multiplying best-response Jacobians (defined in Section 2) through the chain rule while reverse-traversing specific paths of this dataflow graph. This reverse-traversing procedure is crucial for efficiency, as it reduces the computational complexity of our automatic differentiation technique from $\mathcal{O}(d^3)$ to $\mathcal{O}(d^2)$, where $d$ is the dimension of the largest optimization problem in the MLO program.

2. We introduce a software library for MLO, BETTY, built upon the above automatic differentiation technique. Our software design (Section 4), motivated by the dataflow graph interpretation, provides two major benefits: (1) it allows for incorporating various systems support, such as mixed-precision and data-parallel training, for large-scale MLO, and (2) it facilitates implementation of MLO programs of arbitrary complexity while allowing a modular interface for diverse algorithmic and systems design choices. The overall software architecture of BETTY is presented in Figure 1.

3. We empirically demonstrate that BETTY can be used to implement an array of MLO applications with varying scales and complexities (Section 5). Interestingly, we observe that trying out different best-response Jacobian algorithms with our modular interface (which only requires changing one line of code) can lead to up to 11% increase in test accuracy, 14% decrease in GPU memory usage, and 20% decrease in training wall time on various benchmarks, compared with the original papers' implementations. Finally, we showcase the scalability of BETTY to models with hundreds of millions of parameters by performing MLO on the BERT-base model with the help of BETTY's systems support, which was otherwise infeasible.

## 2 BACKGROUND: GRADIENT-BASED MULTILEVEL OPTIMIZATION

To introduce MLO, we first define an important concept known as a "constrained problem" (Vicente & Calamai, 1994).

**Definition 1.** *An optimization problem $P$ is said to be **constrained** by $\lambda$ when its cost function $\mathcal{C}$ has $\lambda$ as an argument in addition to the optimization parameter $\theta$ (i.e. $P : \arg\min_\theta \mathcal{C}(\theta, \lambda, \cdots))$.*

Multilevel optimization (Migdalas et al., 1998) refers to a field of study that aims to solve a nested set of optimization problems defined on a sequence of so-called *levels*, which satisfy two main criteria: **A1)** upper-level problems are constrained by the *optimal* parameters of lower-level problems while **A2)** lower-level problems are constrained by the *nonoptimal* parameters of upper-level problems. Formally, an $n$-level MLO program can be written as:

$$P_n : \quad \theta_n^* = \underset{\theta_n}{\arg\min}\, \mathcal{C}_n(\theta_n, \mathcal{U}_n, \mathcal{L}_n; \mathcal{D}_n) \qquad\qquad \triangleright \text{ Level } n \text{ problem}$$

$$\ddots$$

$$P_k : \qquad\quad \text{s.t. } \theta_k^* = \underset{\theta_k}{\arg\min}\, \mathcal{C}_k(\theta_k, \mathcal{U}_k, \mathcal{L}_k; \mathcal{D}_k) \qquad \triangleright \text{ Level } k \in \{2, \ldots, n-1\} \text{ problem}$$

$$\ddots$$

$$P_1 : \qquad\qquad\quad \text{s.t. } \theta_1^* = \underset{\theta_1}{\arg\min}\, \mathcal{C}_1(\theta_1, \mathcal{U}_1, \mathcal{L}_1; \mathcal{D}_1) \quad \triangleright \text{ Level 1 problem}$$

where, $P_k$ stands for the level $k$ problem, $\theta_k$ / $\theta_k^*$ for corresponding nonoptimal / optimal parameters, and $\mathcal{U}_k$ / $\mathcal{L}_k$ for the sets of constraining parameters from upper / lower level problems. Here, $\mathcal{D}_k$ is the training dataset, and $\mathcal{C}_k$ indicates the cost function. Due to criteria **A1** & **A2**, constraining parameters from upper-level problems should be nonoptimal (*i.e.* $\mathcal{U}_k \subseteq \{\theta_{k+1}, \cdots, \theta_n\}$) while constraining parameters from lower-level problems should be optimal (*i.e.* $\mathcal{L}_k \subseteq \{\theta_1^*, \cdots, \theta_{k-1}^*\}$). Although we denote only one optimization problem per level in the above formulation, each level could in fact have multiple problems. Therefore, we henceforth discard the concept of level, and rather assume that problems $\{P_1, P_2, \cdots, P_n\}$ of a general MLO program are topologically sorted in a "reverse" order (*i.e.* $P_n$ / $P_1$ denote uppermost / lowermost problems).

For example, in hyperparameter optimization formulated as bilevel optimization, hyperparameters and network parameters (weights) correspond to upper and lower level parameters ($\theta_2$ and $\theta_1$). Train / validation losses correspond to $\mathcal{C}_1$ / $\mathcal{C}_2$, and validation loss is dependent on optimal network parameters $\theta_1^*$ obtained given $\theta_2$. Thus, constraining sets for each level are $\mathcal{U}_1 = \{\theta_2\}$ and $\mathcal{L}_2 = \{\theta_1^*\}$.

In this paper, we focus in particular on *gradient-based* MLO, rather than zeroth-order methods like Bayesian optimization (Cui & Bai, 2019), in order to efficiently scale to high-dimensional problems. Essentially, gradient-based MLO calculates gradients of the cost function $\mathcal{C}_k(\theta_k, \mathcal{U}_k, \mathcal{L}_k)$ with respect to the corresponding parameter $\theta_k$, with which gradient descent is performed to solve for optimal parameters $\theta_k^*$ for every problem $P_k$. Since optimal parameters from lower level problems (*i.e.* $\theta_l^* \in \mathcal{L}_k$) can be functions of $\theta_k$ (criterion **A2**), $\frac{d\mathcal{C}_k}{d\theta_k}$ can be expanded using the chain rule as follows:

$$\frac{d\mathcal{C}_k}{d\theta_k} = \underbrace{\frac{\partial \mathcal{C}_k}{\partial \theta_k}}_{\text{direct gradient}} + \sum_{\theta_l^* \in \mathcal{L}_k} \underbrace{\frac{d\theta_l^*}{d\theta_k}}_{\text{best-response Jacobian}} \times \underbrace{\frac{\partial \mathcal{C}_k}{\partial \theta_l^*}}_{\text{direct gradient}} \tag{1}$$

While calculating direct gradients (purple) is straightforward with existing automatic differentiation engines like PyTorch (Paszke et al., 2019), a major difficulty in gradient-based MLO lies in best-response Jacobian[1] (orange) calculation, which will be discussed in depth in Section 3. Once gradient calculation for each level $k$ is enabled via Equation (1), gradient-based optimization is executed from lower to upper level problems in a topologically reverse order, reflecting underlying hierarchies.

## 3 AUTOMATIC DIFFERENTIATION FOR MULTILEVEL OPTIMIZATION

While Equation (1) serves as a mathematical basis for gradient-based multilevel optimization, how to automatically and efficiently carry out such gradient calculation has not been extensively studied

---

[1]We abuse the term Jacobian for a total derivative here while it is originally a matrix of partial derivatives

and incorporated into a software system that can support MLO programs involving many problems with complex dependencies. In this section, we discuss the challenges in building an automatic differentiation library for MLO, and provide solutions to address these challenges.

## 3.1 DATAFLOW GRAPH FOR MULTILEVEL OPTIMIZATION

One may observe that the best-response Jacobian term in Equation (1) is expressed with a *total derivative* instead of a partial derivative. This is because $\theta_k$ can affect $\theta_l^*$ not only through a direct interaction, but also through multiple indirect interactions via other lower-level optimal parameters. For example, consider the four-problem MLO program illustrated in Figure 2. Here, the parameter of Problem 4 ($\theta_{p_4}$) affects the optimal parameter of Problem 3 ($\theta_{p_3}^*$) in two different ways: 1) $\theta_{p_4} \to \theta_{p_3}^*$ and 2) $\theta_{p_4} \to \theta_{p_1}^* \to \theta_{p_3}^*$. In general, we can expand the best-response Jacobian $\frac{d\theta_l^*}{d\theta_k}$ in Equation (1) by applying the chain rule for all paths from $\theta_k$ to $\theta_l^*$ as

$$\frac{d\mathcal{C}_k}{d\theta_k} = \frac{\partial \mathcal{C}_k}{\partial \theta_k} + \sum_{\theta_l^* \in \mathcal{L}_k} \sum_{q \in \mathcal{Q}_{k,l}} \left( \underbrace{\frac{\partial \theta_{q(1)}^*}{\partial \theta_k}}_{\text{upper-to-lower}} \times \left( \prod_{i=1}^{\text{len}(q)-1} \underbrace{\frac{\partial \theta_{q(i+1)}^*}{\partial \theta_{q(i)}^*}}_{\text{lower-to-upper}} \right) \times \frac{\partial \mathcal{C}_k}{\partial \theta_l^*} \right) \tag{2}$$

where $\mathcal{Q}_{k,l}$ is a set of paths from $\theta_k$ to $\theta_l^*$, and $q(i)$ refers to the index of the $i$-th problem in the path $q$ with the last point being $\theta_l^*$. Replacing a total derivative term in Equation (1) with a product of partial derivative terms using the chain rule allows us to ignore indirect interactions between problems, and only deal with direct interactions.

To formalize the path finding problem, we develop a novel dataflow graph for MLO. Unlike traditional dataflow graphs with no predefined hierarchy among nodes, a dataflow graph for multilevel optimization has two different types of directed edges stemming from criteria **A1** & **A2**: *lower-to-upper* and *upper-to-lower*. Each of these directed edges is respectively depicted with green and red arrows in Figure 2. Essentially, a lower-to-upper edge represents the directed dependency between two optimal parameters (*i.e.* $\theta_{P_i}^* \to \theta_{P_j}^*$ with $P_i < P_j$), while an upper-to-lower edge represents the directed dependency between nonoptimal and optimal parameters (*i.e.* $\theta_{P_i} \to \theta_{P_j}^*$ with $P_i > P_j$). Since we need to find paths from the nonoptimal parameter $\theta_k$ to the optimal parameter $\theta_l^*$, the first directed edge must be an upper-to-lower edge (red), which connects $\theta_k$ to some lower-level optimal parameter. Once it reaches the

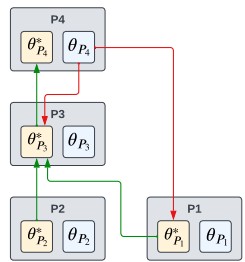

Figure 2: An example dataflow graph for MLO.

optimal parameter, it can only move through optimal parameters via lower-to-upper edges (green) in the dataflow graph. Therefore, every valid path from $\theta_k$ to $\theta_l^*$ will start with an upper-to-lower edge, and then reach the destination only via lower-to-upper edges. The best-response Jacobian term for each edge in the dataflow graph is also marked with the corresponding color in Equation (2). We implement the above path finding mechanism with a modified depth-first search algorithm in BETTY.

## 3.2 GRADIENT CALCULATION WITH BEST-RESPONSE JACOBIANS

Automatic differentiation for MLO can be realized by calculating Equation (2) for each problem $P_k$ ($k = 1, \cdots, n$). However, a naive calculation of Equation (2) could be computationally onerous as it involves multiple matrix multiplications with best-response Jacobians, of which computational complexity is $\mathcal{O}(d^3)$, where $d$ is the dimension of the largest optimization problem in the MLO program. To alleviate this issue, we observe that the rightmost term in Equation (2) is a vector, which allows us to reduce the computational complexity of Equation (2) to $\mathcal{O}(d^2)$ by iteratively performing matrix-vector multiplication from right to left (or, equivalently, reverse-traversing a path $q$ in the dataflow graph). As such, matrix-vector multiplication between the best-response Jacobian and a vector serves as a base operation of efficient automatic differentiation for MLO. Mathematically, this problem can be simply written as follows:

$$\text{Calculate} \quad \frac{\partial w^*(\lambda)}{\partial \lambda} \times v \tag{3}$$

$$\text{Given} \quad w^*(\lambda) = \operatorname*{argmin}_{w} \mathcal{C}(w, \lambda). \tag{4}$$

Two major challenges in the above problems are: 1) approximating the solution of the optimization problem (*i.e.* $w^*(\lambda)$), and 2) differentiating through the (approximated) solution.

In practice, an approximation of $w^*(\lambda)$ is typically achieved by unrolling a small number of gradient steps, which can significantly reduce the computational cost (Franceschi et al., 2017). While we could potentially obtain a better approximation of $w^*(\lambda)$ by running gradient steps until convergence, this procedure alone can take a few days (or even weeks) when the underlying optimization problem is large-scale (Deng et al., 2009; Devlin et al., 2018).

Once $w^*(\lambda)$ is approximated, matrix-vector multiplication between the best-response Jacobian $\frac{dw^*(\lambda)}{d\lambda}$ and a vector $v$ is popularly obtained by either iterative differentiation (ITD) or approximate implicit differentiation (AID) (Grazzi et al., 2020). This problem has been extensively studied in bilevel optimization literature (Finn et al., 2017; Franceschi et al., 2017; Lorraine et al., 2020), and we direct interested readers to the original papers, as studying these algorithms is not the focus of this paper. In BETTY, we provide implementations of several popular ITD/AID algorithms which users can easily plug-and-play for their MLO applications. Currently available algorithms within BETTY include ITD with reverse-mode automatic differentiation (ITD-RMAD) (Finn et al., 2017), AID with Neumann series (AID-NMN) (Lorraine et al., 2020), AID with conjugate gradient (AID-CG) (Rajeswaran et al., 2019), and AID with finite difference (AID-FD) (Liu et al., 2019).

### 3.3 Execution of Multilevel Optimization

In MLO, optimization of each problem should be performed in a topologically reverse order, as the upper-level optimization is constrained by the result of lower-level optimization. To ease an MLO implementation, we also automate such an execution order with the dataflow graph developed in Section 3.1. Specifically, let's assume that there is a lower-to-upper edge between problems $P_i$ and $P_j$ (*i.e.* $\theta_i^* \rightarrow \theta_j^*$). When the optimization process (*i.e.* a small number of gradient steps) of the problem $P_i$ is complete, it can call the problem $P_j$ to start its one-step gradient descent update through the lower-to-upper edge. The problem $P_j$ waits until all lower level problems in $\mathcal{L}_j$ send their calls, and then performs the one-step gradient descent update when all the calls from lower levels are received. Hence, to achieve the full execution of gradient-based MLO, we only need to call the one-step gradient descent processes of the lowermost problems, as the optimization processes of upper problems will be automatically called from lower problems via lower-to-upper edges.

To summarize, automatic differentiation for MLO is accomplished by performing gradient updates of multiple optimization problems in a topologically reverse order based on the lower-to-upper edges (Sec. 3.3), where gradients for each problem are calculated by iteratively multiplying best-response Jacobians obtained with ITD/AID (Sec. 3.2) while reverse-traversing the dataflow graph (Sec. 3.1).

## 4 Software Design

On top of the automatic differentiation technique developed in Section 3, we build an easy-to-use and modular software library, BETTY, with various systems support for large-scale gradient-based MLO. In detail, we break down MLO into two high-level concepts, namely 1) optimization problems and 2) hierarchical dependencies among problems, and design abstract Python classes for both of them. Such abstraction is also motivated by our dataflow graph interpretation, as each of these concepts respectively corresponds to nodes and edges. The architecture of BETTY is shown in Figure 1

**Problem** Each optimization problem $P_k$ in MLO is defined by the parameter (or module) $\theta_k$, the sets of the upper and lower constraining problems $\mathcal{U}_k$ & $\mathcal{L}_k$, the dataset $\mathcal{D}_k$, the cost function $\mathcal{C}_k$, the optimizer, and other optimization configurations (*e.g* best-response Jacobian calculation algorithm, number of unrolling steps). The `Problem` class is an interface where users can provide each of the aforementioned components to define the optimization problem. In detail, each one except for the cost function $\mathcal{C}_k$ and the constraining problems $\mathcal{U}_k$ & $\mathcal{L}_k$ can be provided through the class constructor, while the cost function can be defined through a *"training_step"* method and the constraining problems are automatically provided by `Engine`.

Abstracting an optimization problem by encapsulating module, optimizer, and data loader together additionally allows us to implement various systems support, including mixed-precision, data-parallel training, and gradient accumulation, within the abstract `Problem` class. A similar strategy has also

been adopted in popular frameworks for large-scale deep learning such as DeepSpeed (Rajbhandari et al., 2020). Since implementations of such systems support as well as best-response Jacobian are abstracted away, users can easily plug-and-play different algorithmic and systems design choices, such as unrolling steps or mixed-precision training, via `Config` in a modular fashion. An example usage of `Problem` is shown in Listing 1, and a full list of supported features in `Config` is provided in Appendix F.

```python
1 class MyProblem(Problem):
2     def training_step(self, batch):
3         # Users define the cost function here
4         return cost_fn(batch, self.module, self.other_probs, ...)
5 config = Config(type="darts", unroll_steps=10, fp16=True, gradient_accumulation=4)
6 prob = MyProblem("myproblem", config, module, optimizer, data_loader)
```

Listing 1: `Problem` class example.

**Engine**    While `Problem` manages each optimization problem, `Engine` handles hierarchical dependencies among problems in the dataflow graph. As discussed in Section 3.1, a dataflow graph for MLO has upper-to-lower and lower-to-upper directed edges. We allow users to define two separate graphs, one for each type of edge, using a Python dictionary, in which keys/values respectively represent start/end nodes of the edge. When user-defined dependency graphs are provided, `Engine` compiles them and finds all paths required for automatic differentiation with a modified depth-first search algorithm. Moreover, `Engine` sets constraining problem sets for each problem based on the dependency graphs, as mentioned above. Once all initialization processes are done, users can run a full MLO program by calling `Engine`'s run method, which repeatedly calls the one-step gradient descent procedure of lowermost problems. The example usage of `Engine` is provided in Listing 2.

```python
1 prob1 = MyProblem1(...)
2 prob2 = MyProblem2(...)
3 dependency = {"u2l": {prob1: [prob2]}, "l2u": {prob1: [prob2]}}
4 engine = Engine(problems=[prob1, prob2], dependencies=dependency)
5 engine.run()
```

Listing 2: `Engine` class example.

## 5    EXPERIMENTS

To showcase the general applicability of BETTY, we implement three MLO benchmarks with varying complexities and scales: data reweighting for class imbalance (Sec. 5.1), correcting and reweighting corrupted labels (Sec. 5.2), and domain adaptation for a pretraining/finetuning framework (Sec. 5.3). Furthermore, we analyze the effect of different best-response Jacobian algorithms and system features by reporting GPU memory usage and training wall time. Last but not least, in the Appendix, we include an additional MLO benchmark experiment on differentiable neural architecture search (Appendix A), code examples (Appendix B), training details such as hyperparameters (Appendix C), analyses on various algorithmic and systems design choices (Appendix D and E).

### 5.1    DATA REWEIGHTING FOR CLASS IMBALANCE

Many real-world datasets suffer from class imbalance due to underlying long-tailed data distributions. Meta-Weight-Net (MWN) (Shu et al., 2019) proposes to alleviate the class imbalance issue with a data reweighting scheme where they learn to assign higher/lower weights to data from more rare/common classes. In detail, MWN formulates data reweighting with bilevel optimization as follows:

$$\theta^* = \underset{\theta}{\operatorname{argmin}} \, \mathcal{L}_{val}(w^*(\theta)) \qquad\qquad \triangleright \text{ Reweighting}$$

$$\text{s.t. } w^*(\theta) = \underset{w}{\operatorname{argmin}} \, \frac{1}{N} \sum_{i=1}^{n} \mathcal{R}(L_{train}^i; \theta) \cdot L_{train}^i(f(x_i; w), y_i) \qquad \triangleright \text{ Classification}$$

where $w$ is the network parameters, $L_{train}^i$ is the training loss for the $i$-th training sample, and $\theta$ is the MWN $\mathcal{R}$'s parameters, which reweights each training sample given its training loss $L_{train}^i$.

Following the original paper, we artificially inject class imbalance into the CIFAR-10 dataset by geometrically decreasing the number of data sample for each class, as per an imbalance factor. While

the official implementation, which is built upon Torchmeta (Deleu et al., 2019), only adopts ITD-RMAD for best-response Jacobian calculation, we re-implement MWN with multiple best-response Jacobian algorithms, which only require one-liner changes using BETTY, to study their effect on test accuracy, memory efficiency, and training wall time. The experiment results are given in Table 1.

| | Algorithm | IF 200 | IF 100 | IF 50 | Memory | Time |
|---|---|---|---|---|---|---|
| MWN (original) | ITD-RMAD | 68.91 | 75.21 | 80.06 | 2381MiB | 35.8m |
| MWN (ours, step=1) | ITD-RMAD | 71.96 | 75.13 | 79.50 | 2381MiB | 36.0m |
| MWN (ours, step=1) | AID-CG | 66.23±1.88 | 70.88±1.68 | 75.41±0.61 | 2435MiB | 67.4m |
| MWN (ours, step=1) | AID-NMN | 66.45±1.18 | 70.92±1.35 | 75.90 ±1.73 | 2419MiB | 67.1m |
| MWN (ours, step=1) | AID-FD | 75.45±0.63 | 78.11±0.43 | 81.15±0.25 | **2051MiB** | **28.5m** |
| MWN (ours, step=5) | AID-FD | **76.56**±1.19 | **80.45**±0.73 | **83.11**±0.54 | **2051MiB** | 65.5m |

Table 1: MWN experiment results. IF denotes an imbalance factor. AID-CG/NMN/FD respectively stand for implicit differentiation with conjugate gradient/Neumann series/finite difference.

We observe that different best-Jacobian algorithms lead to vastly different test accuracy, memory efficiency, and training wall time. Interestingly, we notice that AID-FD with unrolling steps of both 1 and 5 consistently achieve better test accuracy (close to SoTA (Tang et al., 2020)) and memory efficiency than other methods. This demonstrates that, while BETTY is developed to support large and general MLO programs, it is still useful for simpler bilevel optimization tasks as well. An additional analysis on the effect of best-response Jacobian can also be found in Appendix D.

Furthermore, to demonstrate the scalability of BETTY to large-scale MLO, we applied MWN to sentence classification with the BERT-base model (Devlin et al., 2018) with 110M parameters. Similarly, we artificially inject class imbalance into the SST dataset, and use AID-FD as our best-response Jacobian calculation algorithm. The experiment results are provided in Table 2.

| | Algorithm | IF 20 | IF 50 | Memory |
|---|---|---|---|---|
| Baseline | AID-FD | 89.99±0.38 | 87.54±0.70 | 8319MiB |
| MWN (fp32) | AID-FD | - | - | Out-of-memory |
| MWN (fp16) | AID-FD | **91.06**±0.09 | **89.79**±0.65 | 10511MiB |

Table 2: MWN+BERT experiment results. fp32 and fp16 respectively stand for full-precision and mixed-precision training.

As shown above, default full-precision training fails due to the CUDA out-of-memory error, while mixed-precision training, which only requires a one-line change in `Config`, avoids this issue while also providing consistent improvements in test accuracy compared to the BERT baseline. This demonstrates that our system features are indeed effective in scaling MLO to large models. We include more analyses on our systems support in Appendix E.

## 5.2 CORRECTING & REWEIGHTING CORRUPTED LABELS

Another common pathology in real-world data science is the issue of label corruption, stemming from noisy data preparation processes (*e.g.* Amazon MTurk). One prominent example of this is in weak supervision (Ratner et al., 2016), where users create labels for large training sets by leveraging multiple weak/noisy labeling sources such as heuristics and knowledge bases. Due to the nature of weak supervision, generated labels are generally noisy, and consequently lead to a significant performance degradation. In this example, we aim to mitigate this issue by 1) correcting and 2) reweighting potentially corrupted labels. More concretely, this problem can be formulated as an *extended* bilevel optimization problem, as, unlike the MWN example, we have two optimization problems—correcting and reweighting—in the upper level, as opposed to one. The mathematical formulation of this MLO program is as follows:

$$\theta^* = \operatorname*{argmin}_{\theta} \mathcal{L}_{val}(w^*(\theta, \alpha)), \quad \alpha^* = \operatorname*{argmin}_{\alpha} \mathcal{L}'_{val}(w^*(\theta, \alpha)) \qquad \triangleright \text{ RWT \& CRT}$$

$$\text{s.t. } w^*(\theta, \alpha) = \operatorname*{argmin}_{w} \frac{1}{N} \sum_{i=1}^{n} \mathcal{R}(L^i_{train}; \theta) \cdot L^i_{train}(f(x_i; w), g(x_i, y_i; \alpha)) \qquad \triangleright \text{ Classification}$$

where, $\alpha$ is the parameter for the label correction network $g$, and $\mathcal{L}'_{val}$ is augmented with the classification loss of the correction network in addition to that of the main classification network $f$ on the clean validation set.

We test our framework on the WRENCH benchmark (Zhang et al., 2021a), which contains multiple weak supervision datasets. In detail, we use a 2-layer MLP as our classifier, AID-FD as our best-response Jacobian algorithm, and Snorkel Data Programming (Ratner et al., 2016) as our weak supervision algorithm for generating training labels. The experiment results are provided in Table 3.

| | TREC | AGNews | IMDB | SemEval | ChemProt | YouTube |
|---|---|---|---|---|---|---|
| Snorkel | 57.52±0.18 | 62.00±0.07 | 71.03±0.55 | 71.00±0.00 | 51.54±0.41 | 77.44±0.22 |
| Baseline | 53.88±1.83 | 80.74±0.20 | 72.26±0.81 | 71.50±0.44 | 54.47±0.78 | 88.16±1.56 |
| +RWT | 57.56±1.41 | 82.79±0.10 | 77.18±0.13 | 77.23±3.38 | 65.33±0.72 | **91.60**±0.75 |
| +RWT&CRT | **66.76**±1.31 | **83.16**±0.20 | **77.80**±0.26 | **84.34**±1.43 | **67.69**±1.17 | 91.52±0.66 |

Table 3: Wrench Results. RWT stands for reweighting and CRT for correction

We observe that simultaneously applying label correction and reweighting significantly improves the test accuracy over the baseline and the reweighting-only scheme in almost all tasks. Thanks to BETTY, adding label correction in the upper-level on top of the existing reweighting scheme only requires defining one more `Problem` class, and accordingly updating the problem dependency in `Engine` (code examples can be found in Appendix B).

## 5.3 DOMAIN ADAPTATION FOR PRETRAINING & FINETUNING

Pretraining/finetuning paradigms are increasingly adopted with recent advances in self-supervised learning (Devlin et al., 2018; He et al., 2020). However, the data for pretraining are oftentimes from a different distribution than the data for finetuning, which could potentially cause negative transfer. Thus, domain adaptation emerges as a natural solution to mitigate this issue. As a domain adaptation strategy, (Raghu et al., 2021) proposes to combine data reweighting with a pretraining/finetuning framework to automatically decrease/increase the weight of pretraining samples that cause negative-/positive transfer. In contrast with the above two benchmarks, this problem can be formulated as trilevel optimization as follows:

$$\theta^* = \underset{\theta}{\mathrm{argmin}}\ \mathcal{L}_{FT}(v^*(w^*(\theta))) \qquad \qquad \triangleright \text{ Reweighting}$$

$$\text{s.t. } v^*(w^*(\theta)) = \underset{v}{\mathrm{argmin}}\ \left(\mathcal{L}_{FT}(v) + \lambda\|v - w^*(\theta)\|_2^2\right) \qquad \triangleright \text{ Finetuning}$$

$$w^*(\theta) = \underset{w}{\mathrm{argmin}}\ \frac{1}{N}\sum_{i=1}^{n}\mathcal{R}(x_i;\theta)\cdot L_{PT}^i(w) \qquad \qquad \triangleright \text{ Pretraining}$$

where $x_i$ / $L_{PT}^i$ stands for the $i$-th pretraining sample/loss, $\mathcal{R}$ for networks that reweight importance for each pretraining sample $x_i$, and $\lambda$ for the proximal regularization parameter. Additionally, $w$, $v$, and $\theta$ are respectively parameters for pretraining, finetuning, and reweighting networks.

We conduct an experiment on the OfficeHome dataset (Venkateswara et al., 2017) that consists of 15,500 images from 65 classes and 4 domains: Art (Ar), Clipart (Cl), Product (Pr), and Real World (RW). Specifically, we randomly choose 2 domains and use one of them as a pretraining task and the other as a finetuning task. ResNet-18 (He et al., 2016) is used for all pretraining/finetuning/reweighting networks, and AID-FT with an unrolling step of 1 is used as our best-response Jacobian algorithm. Following (Bai et al., 2021), the finetuning and the reweighting stages share the same training dataset. We adopted a normal pretraining/finetuning framework without the reweighting stage as our baseline, and the result is presented in Table 4.

Our trilevel optimization framework achieves consistent improvements over the baseline for every task combination at the cost of additional memory usage and wall time, which demonstrates the empirical usefulness of multilevel optimization beyond a two-level hierarchy. Finally, we provide an example of (a simplified version of) the code for this experiment in Appendix B to showcase the usability of our library for a general MLO program.

|  | Algorithm | Cl→Ar | Ar→Pr | Pr→Rw | Rw→Cl | Memory | Time |
|---|---|---|---|---|---|---|---|
| Baseline | N/A | 65.43±0.36 | 87.62±0.33 | 77.43±0.41 | 68.76±0.13 | **3.8GiB** | **290s** |
| + RWT | AID-FD | **67.76**±0.83 | **88.53**±0.42 | **78.58**±0.17 | **69.75**±0.43 | 8.2GiB | 869s |

Table 4: Domain Adaptation for Pretraining & Finetuning results. Reported numbers are classification accuracy on the target domain (right of arrow), after pretraining on the source domain (left of arrow). We note that *Baseline* is a two-layer, and *Baseline + Reweight* a three-layer, MLO program.

## 6  RELATED WORK

**Bilevel & Multilevel Optimization**   There are a myriad of machine learning applications that are built upon bilevel optimization (BLO), the simplest case of multilevel optimization with a two-level hierarchy. For example, neural architecture search (Liu et al., 2019; Zhang et al., 2021b), hyperparameter optimization (Franceschi et al., 2017; Lorraine et al., 2020; Maclaurin et al., 2015), reinforcement learning (Hong et al., 2020; Konda & Tsitsiklis, 1999), data valuation (Ren et al., 2020; Wang et al., 2020), meta learning (Finn et al., 2017; Rajeswaran et al., 2019), and label correction (Zheng et al., 2019) are formulated as BLO. In addition to applying BLO to machine learning tasks, a variety of optimization techniques (Couellan & Wang, 2016; Grazzi et al., 2020; Ji et al., 2021; Liu et al., 2021) have been developed for solving BLO.

Following the popularity of BLO, MLO with more than a two-level hierarchy has also attracted increasing attention recently (Raghu et al., 2021; Somayajula et al., 2022; Such et al., 2020; Xie & Du, 2022). In general, these works construct complex multi-stage ML pipelines, and optimize the pipelines in an end-to-end fashion with MLO. For instance, (Garg et al., 2022) constructs the pipeline of (data generation)–(architecture search)–(classification) and (He et al., 2021) of (data reweighting)–(finetuning)–(pretraining), all of which are solved with MLO. Furthermore, (Sato et al., 2021) study gradient-based methods for solving MLO with theoretical guarantees.

**Multilevel Optimization Software**   There are several software libraries that are frequently used for implementing MLO programs. Most notably, *JAXopt* (Blondel et al., 2021) proposes an efficient and modular approach for AID by leveraging JAX's native autodiff of the optimality conditions. Despite its easy-to-use programming interface for AID, it fails to support combining the chain rule with AID as in Equation (2), because it overrides the default behavior of JAX's automatic differentiation, which takes care of the chain rule. Therefore, it cannot be used for implementing MLO beyond a two-level hierarchy without major changes in the source code and the software design. Alternatively, *higher* (Grefenstette et al., 2019) provides two major primitives of making 1) stateful PyTorch modules stateless and 2) PyTorch optimizers differentiable to ease the implementation of AID/ITD. However, users still need to manually implement complicated internal mechanisms of these algorithms as well as the chain rule with the provided primitives. *Torchmeta* (Deleu et al., 2019) also provides similar functionalities as *higher*, but it requires users to use its own stateless modules implemented in the library rather than patching general modules as in *higher*. Thus, it lacks the support for user's custom modules, limiting its applicability. *learn2learn* (Arnold et al., 2020) focuses on supporting meta learning. However, since meta-learning is strictly a bilevel problem, extending it beyond a two-level hierarchy is not straightforward. Finally, most existing libraries do not have systems support, such as data-parallel training, that could mitigate memory/compute bottlenecks.

## 7  CONCLUSION

In this paper, we aimed to help establish both mathematical and systems foundations for automatic differentiation in MLO. To this end, we devised a novel dataflow graph for MLO, upon which an automatic differentiation procedure is built, and additionally introduced BETTY, a software library with various systems support, that allows for easy programming of a wide range of MLO applications in a modular fashion. We showed that BETTY allows for scaling up to both larger models with many parameters, as well as to MLO programs with multiple dependent problems. As future work, we plan to extend BETTY to support additional algorithmic and systems features, such as best-response Jacobian algorithms for non-differentiable processes, and advanced memory optimization techniques like model-parallel training and CPU-offloading.

ETHICS STATEMENT

Multilevel optimization has the power to be a double-edged sword that can have both positive and negative societal impacts. For example, both 1) defense or attack in an adversarial game, and 2) decreasing or increasing bias in machine learning models, can all be formulated as MLO programs, depending on the goal of the uppermost optimization problem, which is defined by users. Thus, research in preventing malicious use cases of MLO is of high importance.

REPRODUCIBILITY STATEMENT

As one of main contributions of this work is a new software library for scalable multilevel optimization, all of the source code for the library and examples will be released open source with an Apache-2.0 License, including a full implementation of all MLO programs and experiments described in this paper. In addition, for reviewing purposes, we include our source code and easily runnable scripts for all experiments in the supplemental material of this submission.

ACKNOWLEDGEMENTS

We thank all the reviewers for invaluable comments and feedback. EX acknowledges the support of NSF IIS1563887, NSF CCF1629559, NSF IIS1617583, NGA HM04762010002, NIGMS R01GM140467, NSF IIS1955532, NSF CNS2008248, NSF IIS2123952, and NSF BCS2040381. WN was supported in part by NSF (1651565), AFOSR (FA95501910024), ARO (W911NF-21-1-0125), CZ Biohub, Sloan Fellowship, and U.S. Department of Energy Office of Science under Contract No. DE-AC02-76SF00515.

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

# A   ADDITIONAL MULTILEVEL OPTIMIZATION BENCHMARKS

## A.1   DIFFERENTIABLE NEURAL ARCHITECTURE SEARCH

A neural network architecture plays a significant role in deep learning research. However, the search space of neural architectures is so large that manual search is almost impossible. To overcome this issue, DARTS (Liu et al., 2019) proposes an efficient gradient-based neural architecture search method based on the bilevel optimization formulation:

$$\alpha^* = \underset{\alpha}{\text{argmin}} \; \mathcal{L}_{val}(w^*(\alpha), \alpha) \qquad\qquad \triangleright \text{ Architecture Search}$$

$$\text{s.t. } w^*(\alpha) = \underset{w}{\text{argmin}} \; \mathcal{L}_{train}(w; \alpha) \qquad\qquad \triangleright \text{ Classification}$$

where $\alpha$ is the architecture weight and $w$ is the network weight. The original paper uses implicit differentiation with finite difference as its best-response Jacobian algorithm to solve the above MLO program.

We follow the training configurations from the original paper's CIFAR-10 experiment, with a few minor changes. While the original paper performs a finite difference method on the initial network weights, we perform it on the unrolled network weights. This is because we view their best-response Jacobian calculation from the implicit differentiation perspective, where the second-order derivative is calculated based on the unrolled weight. This allows us to unroll the lower-level optimization for more than one step as opposed to strict one-step unrolled gradient descent of the original paper. A similar idea was also proposed in iDARTS (Zhang et al., 2021b). Specifically, we re-implement DARTS with implicit differentiation and finite difference using 1 and 3 unrolling steps. The results are provided in Table 5.

| | Algorithm | Test Acc. | Parameters | Memory | Wall Time |
|---|---|---|---|---|---|
| Random Search | Random | 96.71% | **3.2M** | N/A | N/A |
| DARTS (original) | AID-FD* | 97.24% | 3.3M | 10493MiB | 25.4h |
| DARTS (ours, step=1) | AID-FD | **97.39%** | 3.8M | **10485MiB** | **23.6h** |
| DARTS (ours, step=3) | AID-FD | 97.22% | **3.2M** | **10485MiB** | 28.5h |

Table 5: DARTS re-implementation results. AID-FD refers to implicit differentiation with a finite difference method, and * indicates the difference in the implementation of AID-FD explained above.

Our re-implementation with different unrolling steps achieves a similar performance as the original paper. We also notice that our re-implementation achieves slightly less GPU memory usage and wall time. This is because the original implementation calculates gradients for the architecture weights (upper-level parameters) while running lower-level optimization, while ours only calculates gradients of the parameters for the corresponding optimization stage.

## A.2  CORRECTING & REWEIGHTING CORRUPTED LABELS (EXTENDED)

To further demonstrate the general applicability of BETTY to different datasets and scales, we performed experiments from Section 5.2 in two additional settings.

**Clothing-1M + ResNet-50**  Clothing-1M (Xiao et al., 2015) is a real-world noisy dataset that consists of 1 million fashion images collected from various online shopping websites and has the approximate noise ratio of 38.5%. Following the standard, we use ResNet-50 as our backbone model and attempt to correct and reweight noisy labels with extended bilevel optimization. The experiment result is presented in Table 6

|  | Test Accuracy |
|---|---|
| Baseline | 70.76% |
| +RWT | 75.57% |
| +RWT&CRT | **76.34**% |

Table 6: Clothing-1M + ResNet-50 results.

In this experiment, we are able to empirically show that the MLO application implemented with BETTY works well with a large-scale dataset.

**Wrench + BERT-base**  In recent years, finetuning the pretrained large language model has become the standard for text classification. As the Wrench benchmark mostly consists of text classification datasets, we further applied our "correcting and reweighting corrupted labels" framework to the BERT-base model.

|  | TREC | AGNews | IMDB | SemEval | ChemProt | YouTube |
|---|---|---|---|---|---|---|
| Baseline | 64.14±6.56 | 86.12±0.17 | 71.66±2.05 | 79.93±1.53 | 52.35±0.56 | 93.20±1.44 |
| +RWT | 84.07±4.42 | 89.62±0.60 | **87.85**±0.24 | 87.45±0.69 | 71.42±1.50 | 94.67±0.46 |
| +RWT&CRT | **93.07**±0.31 | **90.40**±0.16 | 87.45±0.39 | **87.92**±0.04 | **75.27**±1.23 | **94.80**±0.80 |

Table 7: Wrench + BERT-base results.

In this experiment, we are able to empirically show that the MLO application implemented with BETTY works well with a large model.

# B  CODE EXAMPLE

Here, we provide simplified code for our experiments from Section 5. Note that every experiment shares a similar code structure when implemented with BETTY.

## B.1  DATA REWEIGHTING FOR CLASS IMBALANCE

```python
train_loader, valid_loader = setup_dataloader()
rwt_module, rwt_optimizer = setup_reweight()
cls_module, cls_optimizer, cls_scheduler = setup_classifier()

# Level 2
class Reweight(ImplicitProblem):
    def training_step(self, batch):
        inputs, labels = batch
        outputs = self.classifier(inputs)
        return F.cross_entropy(outputs, labels)

# Level 1
class Classifier(ImplicitProblem):
    def training_step(self, batch):
        inputs, labels = batch
        outputs = self.module(inputs)
        loss = F.cross_entropy(outputs, labels, reduction="none")
        loss_reshape = torch.reshape(loss, (-1, 1))
        # Reweighting
        weight = self.reweight(loss_reshape.detach())
        return torch.mean(weight * loss_reshape)

upper_config = Config(type="darts", retain_graph=True)
lower_config = Config(type="default", unroll_steps=5)

reweight = Reweight(name="reweight",
                    config=upper_config,
                    module=rwt_module,
                    optimizer=rwt_optimizer,
                    train_data_loader=valid_loader)
classifier = Classifier(name="classifier",
                        config=lower_config,
                        module=cls_module,
                        optimizer=cls_optimizer,
                        scheduler=cls_scheduler,
                        train_data_loader=train_loader)

probs = [reweight, classifier]
u2l = {reweight: [classifier]}
l2u = {classifier: [reweight]}
depends = {"l2u": l2u, "u2l": u2l}

engine = Engine(problems=probs, dependencies=depends)
engine.run()
```

Listing 3: Simplified code of "Data Reweighting for Class Imbalance"

## B.2 CORRECTING & REWEIGHTING CORRUPTED LABELS

```
1  train_loader, valid_loader = setup_dataloader()
2  rwt_module, rwt_optimizer = setup_reweight()
3  crt_module, crt_optimizer = setup_correct()
4  cls_module, cls_optimizer, cls_scheduler = setup_classifier()
5
6  # Level 2
7  class Correct(ImplicitProblem):
8      def training_step(self, batch):
9          inputs, labels = batch
10         outputs, embeds = self.classifier(inputs, return_embeds=True)
11         correct_outputs = self.module(embeds, test=True)
12         ce_loss = F.cross_entropy(outputs, labels)
13         aux_loss = F.cross_entropy(correct_outputs, labels)
14         return ce_loss + aux_loss
15
16 # Level 2
17 class Reweight(ImplicitProblem):
18     def training_step(self, batch):
19         inputs, labels = batch
20         outputs = self.classifier(inputs)
21         return F.cross_entropy(outputs, labels)
22
23 # Level 1
24 class Classifier(ImplicitProblem):
25     def training_step(self, batch):
26         inputs, labels = batch
27         outputs, embeds = self.module(inputs, return_embeds=True)
28         # Correcting
29         new_labels = self.correct(embeds, labels)
30         log_softmax = F.log_softmax(outputs, dim=-1)
31         loss = torch.sum(-log_softmax * new_labels, dim=-1)
32         loss_reshape = torch.reshape(loss, (-1, 1))
33         # Reweighting
34         weight = self.reweight(loss_reshape.detach())
35         return torch.mean(weight * loss_reshape)
36
37 upper_config = Config(type="darts", retain_graph=True)
38 lower_config = Config(type="default", unroll_steps=5)
39
40 correct = Correct(name="correct",
41                   config=upper_config,
42                   module=crt_module,
43                   optimizer=crt_optimizer,
44                   train_data_loader=valid_loader)
45 reweight = Reweight(name="reweight",
46                     config=upper_config,
47                     module=rwt_module,
48                     optimizer=rwt_optimizer,
49                     train_data_loader=valid_loader)
50 classifier = Classifier(name="classifier",
51                         config=lower_config,
52                         module=cls_module,
53                         optimizer=cls_optimizer,
54                         scheduler=cls_scheduler,
55                         train_data_loader=train_loader)
56
57 probs = [correct, reweight, classifier]
58 u2l = {correct: [classifier], reweight: [classifier]}
59 l2u = {classifier: [correct, reweight]}
60 depends = {"l2u": l2u, "u2l": u2l}
61
62 engine = Engine(problems=probs, dependencies=depends)
63 engine.run()
```

Listing 4: Simplified code of "Correcting & Reweighting Corrupted Labels"

### B.3 DOMAIN ADAPTATION FOR PRETRAINING & FINETUNING

```python
1  # Get module, optimizer, lr_scheduler, data loader for each problem
2  pt_module, pt_optimizer, pt_scheduler, pt_loader = setup_pretrain()
3  ft_module, ft_optimizer, ft_scheduler, ft_loader = setup_finetune()
4  rw_module, rw_optimizer, rw_scheduler, rw_loader = setup_reweight()
5
6  # Level 1
7  class Pretrain(ImplicitProblem):
8      def training_step(self, batch):
9          inputs, targets = batch
10         outs = self.module(inputs)
11         loss_raw = F.cross_entropy(outs, targets, reduction="none")
12
13         logit = self.reweight(inputs)
14         weight = torch.sigmoid(logit)
15         return torch.mean(loss_raw * weight)
16
17  # Level 2
18  class Finetune(ImplicitProblem):
19      def training_step(self, batch):
20          inputs, targets = batch
21          outs = self.module(inputs)
22          loss = F.cross_entropy(outs, targets, reduction="none")
23          loss = torch.mean(ce_loss)
24          # Proximal regularization
25          for (n1, p1), p2 in zip(self.module.named_parameters(), self.
       pretrain.module.parameters()):
26              lam = 0 if "fc" in n1 else args.lam
27              loss += lam * (p1 - p2).pow(2).sum()
28          return loss
29
30  # Level 3
31  class Reweight(ImplicitProblem):
32      def training_step(self, batch):
33          inputs, targets = batch
34          outs = self.finetune(inputs)
35          return F.cross_entropy(outs, targets)
36
37  # Define optimization configurations
38  reweight_config = Config(type="darts", step=1, retain_graph=True)
39  finetune_config = Config(type="default", step=1)
40  pretrain_config = Config(type="default", step=1)
41
42  pretrain = Pretrain("pretrain", pt_config, pt_module, pt_optimizer
43                      pt_scheduler, pt_loader)
44  finetune = Finetune("finetune", ft_config, ft_module, ft_optimizer
45                      ft_scheduler, ft_loader)
46  reweight = Reweight("reweight", rw_config, rw_module, rw_optimizer
47                      rw_scheduler, rw_loader)
48
49  probs = [reweight, finetune, pretrain]
50  u2l = {reweight: [pretrain]}
51  l2u = {pretrain: [finetune], finetune: [reweight]}
52  depends = {"u2l": u2l, "l2u": l2u}
53  engine = Engine(problems=probs, dependencies=depends)
54  engine.run()
```

Listing 5: Simplified code of "Domain Adaptation for Pretraining & Finetuning"

## B.4 DIFFERENTIABLE NEURAL ARCHITECTURE SEARCH

```python
train_loader, valid_loader = setup_dataloader()
arch_module, arch_optimizer = setup_architecture()
cls_module, cls_optimizer, cls_scheduler = setup_classifier()

# Level 2
class Architecture(ImplicitProblem):
    def training_step(self, batch):
        x, target = batch
        alphas = self.module()
        return self.classifier.module.loss(x, alphas, target)

# Level 1
class Classifier(ImplicitProblem):
    def training_step(self, batch):
        x, target = batch
        alphas = self.architecture()
        return self.module.loss(x, alphas, target)

arch_config = Config(type="darts",
                     step=1,
                     retain_graph=True,
                     first_order=True)
cls_config = Config(type="default")

architecture = Architecture(name="architecture",
                            config=arch_config,
                            module=arch_module,
                            optimizer=arch_optimizer,
                            train_data_loader=valid_loader)
classifier = Classifier(name="classifier",
                        config=cls_config,
                        module=cls_module,
                        optimizer=cls_optimizer,
                        scheduler=cls_scheduler,
                        train_data_loader=train_loader)

probs = [architecture, classifier]
u2l = {architecture: [classifier]}
l2u = {classifier: [architecture]}
depends = {"l2u": l2u, "u2l": u2l}

engine = Engine(problems=probs, dependencies=depends)
engine.run()
```

Listing 6: Simplified code of "Differentiable Neural Architecture Search"

## C    EXPERIMENT DETAILS

In this section, we provide further training details (*e.g.* hyperparameters) of each experiment.

### C.1    DATA REWEIGHTING FOR CLASS IMBALANCE

**Dataset**    We reuse the long-tailed CIFAR-10 dataset from the original paper (Shu et al., 2019) as our inner-level training dataset. More specifically, the imbalance factor is defined as the ratio between the number of training samples from the most common class and the most rare class. The number of training samples of other classes are defined by geometrically interpolating the number of training samples from the most common class and the most rare class. We randomly select 100 samples from the validation set to construct the upper-level (or meta) training dataset, and use the rest of it as the validation dataset, on which classification accuracy is reported in the main text.

**Meta-Weight-Network**    We adopt a MLP with one hidden layer of 100 neurons (*i.e.* 1-100-1) as our Meta-Weight-Network (MWN). It is trained with the Adam optimizer (Kingma & Ba, 2014) whose learning rate is set to 0.00001 throughout the whole training procedure, momentum values to (0.9, 0.999), and weight decay value to 0. MWN is trained for 10,000 iterations and learning rate is fixed throughout training.

**Classification Network**    Following the original MWN work (Shu et al., 2019), we use ResNet32 (He et al., 2016) as our classification network. It is trained with the SGD optimizer whose initial learning rate is set to 0.1, momentum value to 0.9, and weight decay value to 0.0005. Training is performed for 10,000 iterations, and we decay the learning rate by a factor of 10 on the iterations of 5,000 and 7,500.

### C.2    CORRECTING & REWEIGHTING CORRUPTED LABELS

**Dataset**    We directly use TREC, AGNews, IMDB, SemEval, ChemProt, YouTube text classification datasets from the Wrench benchmark (Zhang et al., 2021a). More specifically, we use the training split of each dataset for training the classification network, and the validation split for training the correcting and the reweighting networks. Test accuracy is measured on the test split.

**Correct Network**    Our correct network takes the penultimate activation from the classification network, and outputs soft labels through the linear layer and the softmax layer. These new soft labels are interpolated with the original labels via the reweighting scheme which is achieved with 2-layer MLP. As our reweighting network, the correct network is trained with Adam optimizer whose learning rate is set to 0.00001, momentum values to (0.9, 0.999), and weight decay value to 0.

**Reweighting Network**    For our reweighting network, we reuse Meta-Weight-Net from the "Data Reweighting for Class Imbalance" experiment, follow all the training details.

**Classification Network**    As our classification network, we adopt a 2-layer MLP with the hidden size of 100. The classification network is trained for 30,000 iterations with the SGD optimizer whose learning rate is set to 0.003, momentum to 0.9, and weight decay to 0.0001. Learning rate is decayed to 0 with the cosine annealing schedule during training.

### C.3    DOMAIN ADAPTATION FOR PRETRAINING & FINETUNING

**Dataset**    We split each domain of the OfficeHome dataset (Venkateswara et al., 2017) into training/validation/test datasets with a ratio of 5:3:2. The pretraining network is trained on the training set of the source domain. Finetuning and reweighting networks are both trained on the training set of the target domain following the strategy proposed in (Bai et al., 2021). The final performance is measured by the classification accuracy of the finetuning network on the test dataset of the target domain.

**Pretraining Network**    We use ResNet18 (He et al., 2016) pretrained on the ImageNet dataset (Deng et al., 2009) for our pretraining network. Following the popular transfer learning strategy, we split the network into two parts, namely the feature (or convolutional layer) part and the classifier (or fully-connected layer) part, and each part is trained with different learning rates. Specifically, learning rates for the feature and the classifier parts are respectively set to 0.001 and 0.0001 with the Adam optimizer. They share the same weight decay value of 0.0005 and momentum values of (0.9, 0.999). Furthermore, we encourage the network weight to stay close to the pretrained weight by introducing the additional proximal regularization with the regularization value of 0.001. Training is performed for 1,000 iterations, and the learning rate is decayed by a factor of 10 on the iterations of 400 and 800.

**Finetuning Network**    The same architecture and optimization configurations as the pretraining network are used for the finetuning network. The proximal regularization parameter, which encourages the finetuning network parameter to stay close to the pretraining network parameter, is set to 0.007.

**Reweighting Network**    The same architecture and optimization configurations as the pretraining network are used for the reweighting network, except that no proximal regularization is applied to the reweighting network.

## C.4    DIFFERENTIABLE NEURAL ARCHITECTURE SEARCH

**Dataset**    Follwing the original paper (Liu et al., 2019), we use the first half of the CIFAR-10 training dataset as our inner-level training dataset (*i.e.* classification network) and the other half as the outer-level training dataset (*i.e.* architecture network). Training accuracy reported in the main text is measured on the CIFAR-10 validation dataset.

**Architecture Network**    We adopt the same architecture search space as in the original paper (Liu et al., 2019) with 8 operations, and 7 nodes per convolutional cell. The architecture parameters are initialized to zero to ensure equal softmax values, and trained with the Adam optimizer (Kingma & Ba, 2014) whose learning rate is fixed to 0.0003, momentum values to (0.5, 0.999), and weight decay value to 0.001 throughout training. Training is performed for 50 epochs.

**Classification Network**    Given the above architecture parameters, we set our classification network to have 8 cells and the initial number of channels to be 16. The network is trained with the SGD optimizer whose initial learning rate is set to 0.025, momentum to 0.9, and weight decay value to 0.0003. Training is performed for 50 epochs, and the learning rate is decayed following the cosine annealing schedule without restart to the minimum learning rate of 0.001 by the end of training.

# D DESIGN CHOICE ANALYSIS

In this section, we visually compare the convergence speed of different best-response Jacobian algorithms with the loss convergence graphs on the synthetic hyperparameter optimization task and the data reweighting task (Section 5.1). Specifically, we analyze the convergence speed in terms of both 1) the number of steps and 2) training time, as the per-step computational cost differs for each algorithm.

## D.1 SYNTHETIC HYPERPARAMETER OPTIMIZATION

Following (Grazzi et al., 2020), we constructed a synthetic hyperparameter optimization task where we optimize the weight decay value for *every* parameter in simple binary logistic regression. Mathematically, this problem can be formulated as bilevel optimization as follows:

$$\lambda^* = \arg\min_\lambda \ \text{sigmoid}(y_u x_u^T w^*)$$

$$w^* = \arg\min_w \ \text{sigmoid}(y_l x_l^T w^*) + \frac{1}{2} w^T diag(\lambda) w$$

where, $(x_l, y_l)$ and $(x_u, y_u)$ are repsectively the training datasets for the lower-(and upper-)level problems, with $x \in \mathbb{R}^{n \times d}$ and $y \in \mathbb{R}^{n \times 1}$. Here, $n$ is the number of training data in each dataset and $d$ is the dimension of the feature vector. $w \in \mathbb{R}^{d \times 1}$ is the logistic regression parameter, and $\lambda \in \mathbb{R}^{d \times 1}$ is the hyperparameter (i.e. the per-parameter weight decay value).

Given the above setup, we compared four different best-reponse Jacobian algorithms: 1) ITD-RMAD, 2) AID-FD, 3) AID-CG, and 4) AID-Neumann. For the fair comparison, we fixed the unrolling step to 100 for all algorithms. The experiment result is presented below:

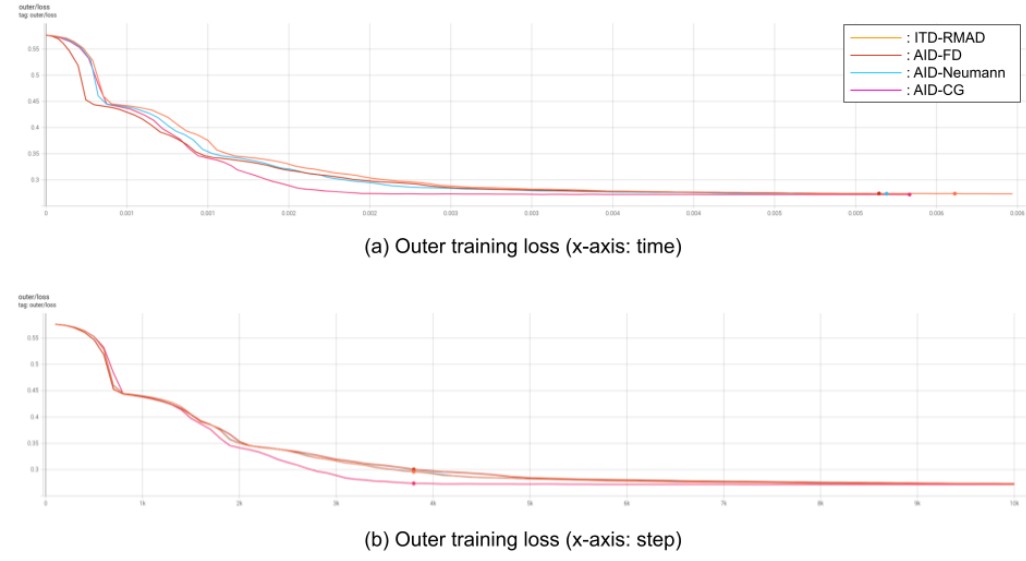

(a) Outer training loss (x-axis: time)

(b) Outer training loss (x-axis: step)

Figure 3: Convergence analysis of different best-response Jacobian algorithms on the synthetic hyperparameter optimization task

As shown in Figure 3, AID-CG achieves the fastest convergence both in terms of training steps and training time. However, AID-FD achieves the fastest per-step computation time as it is the only algorithms that doesn't require the explicit calculation of the second-order derivative (i.e. Hessian).

## D.2 DATA REWEIGHTING

To study how different best-response Jacobian algorithms perform on more complex tasks, we repeated the above experiment on the data reweighting task from Section 5.1. Again, for the fair

comparison, we used the same unrolling step of 1 for all algorithms. The experiment result is provided in Figure 4.

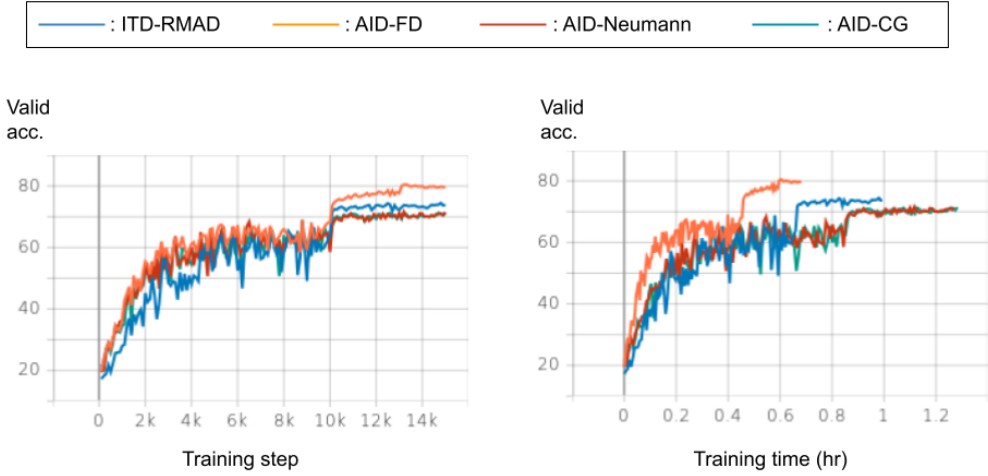

Figure 4: Convergence analysis of different best-response Jacobian algorithms on the data reweighting task

.

Unlike in the synthetic hyperparameter optimization task, AID-FD achieves the fastest convergence in terms of training steps and training time as well as the best final validation accuracy. As AID-FD doesn't require any second-order derivative calculation, it also achieves the minimal per-step computation cost.

Above two experiments follow the no free lunch theorem: the optimal design choice can vary for different tasks without golden rules. However, thanks to the modular interface for switching between different design choices (in `Config`), only minimal programming efforts would be needed with BETTY, expediting the research cycle.

# E   Systems Support

In this section, we perform additional analyses on the memory saving effects of our system features with two benchmarks: (1) differentiable neural architecture search and (2) data reweighting for class imbalance.

## E.1   Differentiable Neural Architecture Search

|  | Baseline | + mixed-precision |
|---|---|---|
| GPU Memory Usage | 9867MiB | **5759MiB** |

Table 8: GPU memory usage analysis for DARTS.

## E.2   Data Reweighting for Class Imbalance

In this experiment, we use ResNet50 (He et al., 2016) instead of ResNet30, to better study the memory reduction from our system features, when the larger model is used. Importantly, we also test the data-parallel training feature in addition to the mixed-precision training feature.

|  | Baseline | + mixed-precision | + data-parallel (2 GPUs) |
|---|---|---|---|
| GPU Memory Usage | 6817MiB | 4397MiB | **3185/3077MiB** (GPU0/1) |

Table 9: GPU memory usage analysis for MWN with ResNet-50.

As shown above, we observe more reduction in memory usage as we add more system features.

# F SUPPORTED FEATURES

Here, we summarize the supported features within BETTY.

| Category | Features |
|---|---|
| Best-response Jacobian algorithms | · ITD-RMAD
· AID-FD
· AID-NMN
· AID-CG |
| Systems | · Mixed-precision
· Data-parallel
· Gradient accumulation |
| Logging | · Default Python logging
· TensorBoard
· Weights & Biases |
| Miscellaneous | · Gradient clipping
· Early stopping |

Table 10: Supported features in BETTY

# G    DATAFLOW GRAPHS FOR EXPERIMENTS

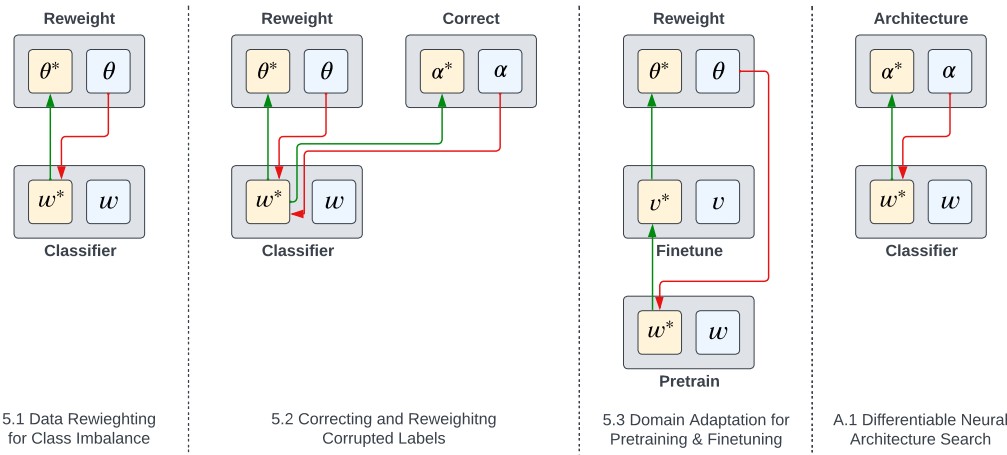

Figure 5: Dataflow graphs for all our experiments

