# OpenReview forum: "Betty: An Automatic Differentiation Library for Multilevel Optimization"
_ICLR.cc/2023/Conference — ICLR 2023 notable top 5%_

### Official Review · Reviewer_G2gf · 2022-10-17

**Confidence:** 3
**Correctness:** 4
**Technical Novelty And Significance:** 4
**Empirical Novelty And Significance:** 2
**Recommendation:** 8

**Clarity, Quality, Novelty And Reproducibility:**

The paper is clear, the proposed library novel, and results should be easily reproducible once the software is available.

**Strength And Weaknesses:**

+ important problem
+ useful library that achieves good results

**Summary Of The Paper:**

The paper proposes Betty, a library for automatic differentiation for
optimization. The authors describe the library and its design and evaluate it on
a number of problems, comparing to other approaches.

**Summary Of The Review:**

The paper is well written, the design of the proposed library makes sense and
allows for more efficient differentiation than other methods, and the
experimental results are convincing. This is a nice paper that should be
accepted.

---

> ### Author Response · Authors · 2022-11-12
> **Response to Reviewer G2gf**
>
> Thank you for the useful  review and positive feedback.
>
> We would like to highlight that we have updated our paper with two additional experiments, which further demonstrate the general applicability of Betty on large-scale datasets (e.g. Clothing-1M) and large models (e.g. BERT-base). These are described in Appendix A.2, and we copy the main results below:
>
>
> ### **Reweighting and Correcting Corrupted Labels w/ Clothing-1M + ResNet-50**
>
> |          | Test Accuracy |
> |----------|---------------|
> | Baseline | 70.76         |
> | +RWT     | 75.57         |
> | +RWT&CRT | **76.34**         |
>
> ### **Reweighting and Correcting Corrupted Labels w/ BERT-base + WRENCH**
>
> |          | TREC        | AGNews      | IMDB        | SemEval     | ChemProt    | YouTube     |
> |----------|-------------|-------------|-------------|-------------|-------------|-------------|
> | Baseline | 64.14+-6.56 | 86.12+-0.17 | 71.66+-2.05 | 79.93+-1.53 | 52.35+-0.56 | 93.20+-1.44 |
> | +RWT     | 84.07+-4.42 | 89.62+-0.60 | **87.85**+-0.24 | 87.45+-0.69 | 71.42+-1.50 | 94.67+-0.46 |
> | +RWT&CRT | **93.07**+-0.31 | **90.40**+-0.16 | 87.45+-0.39 | **87.92**+-0.04 | **75.27**+-1.23 | **94.80**+-0.80 |
>
>
> We hope the above experiments make our work even stronger. If you have any other comments regarding the paper, we are more than happy to discuss them, so feel free to let us know!

---

> > ### Comment · Reviewer_G2gf · 2022-11-28
> > **Thank you**
> >
> > Thank you for the update!

---

### Official Review · Reviewer_5V8s · 2022-10-23

**Confidence:** 3
**Correctness:** 4
**Technical Novelty And Significance:** 3
**Empirical Novelty And Significance:** 3
**Recommendation:** 6

**Clarity, Quality, Novelty And Reproducibility:**

This paper is clearly written and the quality is good. The proposed method is valid and the novelty of this paper is relative good. The experiments in this paper is reproducible and the codes of the software will be made publicly available.

**Details Of Ethics Concerns:**

There are no ethics concerns.

**Strength And Weaknesses:**

Strength:
1. The paper is well written and the presentation is clear.
2. Applications of MLO in machine learning are very common hence studies of more efficient automatic differentiation methods are interesting.
3. The evaluations are performed on several commonly used scenarios. The empirical evidences for the validation of the proposed method is relatively strong.

Weaknesses:
1. The experiments are conducted for MLO with at most three-levels but the general method is applicable to MLO with more levels. Applications with at most three-levels might not demonstrate the full advantages of the proposed method.
2. The proposed method is specifically designed for deep learning based classification problems. It's unclear whether the proposed method is generally applicable for problems in domains outside machine learning.
3. More comparisons with existing software libraries should be considered other than the single default baseline in each application.
4. Experiments are mainly focused on the relatively simple CIFAR-10 benchmark.

**Summary Of The Paper:**

This paper proposed an automatic differentiation method for multilevel optimization (MLO) by a special dataflow graph. Specifically, by reverse-traversing the paths of the dataflow graph, the best-response Jacobians can be computed iteratively following the chain rule. This method can reduce the computational complexity of automatic differentiation from O(d^3) to O(d^2). Based on the proposed automatic differentiation method, this paper has introduced a software library that supports mixed-precision and data-parallel training. Finally, this paper demonstrated the library on several MLO applications and observed higher accuracy than baselines.

**Summary Of The Review:**

The technique proposed in this paper is valid and show potentials to be applied in common MLO applications. However, the evaluations of the proposed method can be made more extensive by considering wider applications and more baseline methods. Also, it's good to show experimentally the improvement of computational efficiency over baseline methods.

---

> ### Author Response · Authors · 2022-11-12
> **Response to Reviewer 5V8s**
>
> We want to thank the reviewer for the thoughtful review and constructive feedback. In this response, we address each of your comments, and clarify some of misunderstandings.
>
> We would first like to highlight that we have updated our paper with two additional experiments, which further demonstrate the general applicability of Betty on large-scale datasets (e.g. Clothing-1M) and large models (e.g. BERT-base). These are described in Appendix A.2, and we copy the main results below:
>
> ### **Reweighting and Correcting Corrupted Labels w/ Clothing-1M + ResNet-50**
>
> |       | Test Accuracy |
> |----|-----|
> | Baseline | 70.76   |
> | +RWT     | 75.57  |
> | +RWT&CRT | **76.34**  |
>
>
> ### **Reweighting and Correcting Corrupted Labels w/ BERT-base + Wrench**
>
> |        | TREC      | AGNews    | IMDB   | SemEval    | ChemProt   | YouTube   |
> |-------|-------|-------|-------|-------|-------|-------|
> | Baseline | 64.14+-6.56 | 86.12+-0.17 | 71.66+-2.05 | 79.93+-1.53 | 52.35+-0.56 | 93.20+-1.44 |
> | +RWT     | 84.07+-4.42 | 89.62+-0.60 | **87.85**+-0.24 | 87.45+-0.69 | 71.42+-1.50 | 94.67+-0.46 |
> | +RWT&CRT | **93.07**+-0.31 | **90.40**+-0.16 | 87.45+-0.39 | **87.92**+-0.04 | **75.27**+-1.23 | **94.80**+-0.80 |
>
> ### **Questions**
>
> > **Q1** The experiments are conducted for MLO with at most three-levels but the general method is applicable to MLO with more levels. Applications with at most three-levels might not demonstrate the full advantages of the proposed method.
>
> **A1** We are glad that the reviewer recognized our library is generally applicable to MLO with arbitrary structures. We noticed that currently available *strong and interesting* MLO applications are mostly two-level or three-levels (sometimes with multiple problems per level), and therefore decided to also focus on those applications in this paper. To the best of our knowledge, there has been no existing work that explicitly demonstrates the seamless support for even these problem structures that we show in our experiments, such as two-level optimization with multiple parent problems (Sec 5.2) or three-level optimization problems (Sec 5.3).
>
> > **Q2** The proposed method is specifically designed for deep learning based classification problems. It's unclear whether the proposed method is generally applicable for problems in domains outside machine learning.
>
> **A2** Betty should be applicable to any machine learning problems that can be solved with gradient-based optimization, such as linear regression and SVM. In fact, the synthetic hyperparameter optimization experiment in Appendix D.1 adopts logistic regression instead of deep learning. However, given the limited space, we wanted to mainly focus on deep learning based classification examples in our manuscript as they are some of the most common machine learning problems.
>
> > **Q3** More comparisons with existing software libraries should be considered other than the single default baseline in each application.
>
> **A3** Existing software libraries focus on only very specific aspects or applications of MLO (e.g. model-agnostic meta-learning), and are not able to support the problems/experiments that we focus on in our paper without major changes to their source code. Likewise, the baseline (or official) implementations of the applications in our paper also chose very specific libraries based on the features that they needed. We therefore didn’t have many options other than adopting the default/official baseline for each application. We consider one of the major advantages of Betty is that it can be used to implement a wide range of MLO applications in a unified and standardized way. We hope our library can allow for systematic research in MLO, which has been largely fragmented to date.
>
> > **Q4** Experiments are mainly focused on the relatively simple CIFAR-10 benchmark.
>
> **A4** We want to kindly point out that we use a number of other datasets/benchmarks in our experiments (beyond CIFAR-10). For example, our large-scale “Data Reweighting for Class Imbalance” experiment is performed on the *SST benchmark* with the BERT-base model. Furthermore, we adopted the *WRENCH benchmark with 6 different datasets* for our “Correcting and Reweighting Corrupted Labels” experiment. Finally, the “Domain Adaptation for Pretraining & Finetuning” application used the *OfficeHome benchmark with 4 different sub-datasets*.
>
> Additionally, as mentioned above, we have now added two additional experiments to our paper, including a large-scale Clothing-1M dataset, for additional variety.
>
> > **Q5** Also, it's good to show experimentally the improvement of computational efficiency over baseline methods.
>
> **A5** We kindly want to point out that, in a number of experiments, we have showcased the computational/memory efficiency of our library, e.g. in Tables 1, 8, and 9.
>
> We hope our rebuttal clarified most of your concerns regarding our paper. If you have any additional comments, we are more than happy to discuss them. Thanks again for your review!

---

> ### Author Response · Authors · 2022-12-06
> **Response to Reviewer 5V8s – Follow-Up**
>
> We wanted to thank you again for the insightful suggestions, and kindly ask if you could please let us know if our answers are satisfactory or further elaboration is needed.

---

### Official Review · Reviewer_MAHC · 2022-10-24

**Confidence:** 4
**Correctness:** 3
**Technical Novelty And Significance:** 3
**Empirical Novelty And Significance:** 4
**Recommendation:** 10

**Clarity, Quality, Novelty And Reproducibility:**

Clarity
---------
The paper reads well overall, and describes multi-level optimization and the approach taken in understandable terms.

Some things could be improved a bit, or clarified, in particular:
1. It would be nice to have diagrams similar to Fig. 2 for the actual examples, especially 5.3 / B.3
2. The direction of arrows / dependencies is not obvious to grasp, when they're presented as "dependencies". They may plausibly be "use -> definition" as well as "definition -> use"
3. "training_step" is a confusing name when what seems to be described is a cost function, rather than an update step.
4. How does it work when there is more than 1 top-level problem, as in 5.2 / B.2? Are both problems implicitly given a weight? It looks like there would usually be a trade-off.
5. It would probably help if indices i, j, k, l were used in a more consistent way (in which indices are upper vs. lower, for instance)

Quality
----------
The paper is convincing, the use cases described in the text (then in appendices) demonstrate simple examples (bilevel optimization) as well as non-trivial ones (5.3, where dependencies would otherwise form a cycle).
Experiments reproduce (or improve on) existing problems, showcasing the usefulness (and correctness) of the library.

Novelty
----------
To my knowledge, this is the first library that explicitly organizes dependencies in MLO in such a way, enabling to easily express MLO problems without explicitly redefining gradient expressions around solvers multiple times.

The factorization of best-response-Jacobian * vector products (Section 3.2) is similar to the usual factorization of Jacobian-vector products in forward-mode AD, so I don't think that part is really novel, but it's not presented as one major result, so that's OK.

Reproducibility
--------------------
The provided code and examples are great to help with reproducibilitiy of the concept, and specific experiments.
I have no doubt the results can be reproduced.

Other questions
----------------------
Definition 1 defines a "constrained" optimization problem, but that is quite different from other usual ways constraints can be defined in optimization problems, for instance equality and unequality constraints on the parameters.
I suppose equality constraints can be expressed explicitly with slack variables in BETTY (although it may be tedious to write, if there is no specific support), but what about inequality constraints? Are there any incompatibilities in the way problems are expressed? Could they be supported by a future version of the framework?

Minor & typos
1. On p.3, when defining $P_1$, should the last term of the argmin be $\mathcal D_1$ instead of $\mathcal D_k$? Or maybe $\emptyset$?
2. In B.2 (p. 15), both `Correct` and `Reweight` have exactly the same implementation for `training_step`, but the text and math in 5.2 (p.7-8) mention a different loss, $\mathcal L'_{val}$, "augmented with the classification loss of the correction network", which I don't see in the code listing.


**Strength And Weaknesses:**

Strengths
-------------
1. Principled expansion of bilevel optimization (which enjoy increasing popularity) to a DAG of sub-problems, well described
2. Understandable and usable formalism to define such problems in practice in a framework
3. Several informative examples, demonstrating the range of capabilities of the framework
4. Comparison with (and exploration around) SOTA, demonstrating similar (or improved) results and efficiency.
5. Extensive appendix with details
6. Opensource code

Weaknesses
------------------
1. Clarity could be improved in some places, as it can be a difficult problem to grasp (see below)


**Summary Of The Paper:**

The paper provides a representation for multilevel optimization problems, going beyond the more common bilevel optimization problems, as a dataflow graph. This graph contains two type of edges, representing 1) dependencies of higher-level optimal parameters on lower-level optimal parameters, and 2) dependencies of lower-level optimal parameters on higher-level non-optimal parameters. It can be used to efficiently compute best-response Jacobians and total gradients.
It then introduces a software library based on that principle, which enables user to express multi-level optimization problems in short programs, by defining each problem's cost function and solver configuration, and dependencies between problems.
Experiments showcase the computational (and memory) efficiency of the library, as well as its flexibility, on existing problems and variants.

**Summary Of The Review:**

This paper introduces a well-motivated framework to express and solve multilevel optimization problems. The paper is quite clear, and the experiments are convincing regarding the correctness, ease-of-use, and efficiency of the implementation. Exploring the MLO space beyond bi-level (or even stacked) optimization problems is of interest to the community, and this frameworks seems to be the first in that domain, and may enable it.
I recommend to accept it.

**Update after response**
Thanks to the authors for their answers and for elaborating on the points raised. I still recommend acceptance, and maintain my rating.

---

> ### Author Response · Authors · 2022-11-12
> **Response to Reviewer MAHC 2/2**
>
> ### **Other Questions**
> > **Q6** Terminology regarding “Constraints”.
>
> **A6** We totally understand your points on this issue. Throughout writing the paper, we generally had a hard time choosing the right terminology here, as the field is largely fragmented (the same problem is given a variety of names such as multilevel optimization, Stackelberg game, or just meta-learning, each of which uses different terms). If you view the optimization problem from the probabilistic perspective, maybe “conditioned” makes more sense than “constrained”.
>
> Regarding the inequality constraints, we believe you can turn most inequality constraints into equality constraints by introducing slack variables as is common in the Lagrange Multiplier Approach for the constrained optimization problem. More specifically, an inequality constraint $f(x) >= 0$ can be turned into an equality constraint with the slack variable $s$ as $f(x) - s^2 = 0$.
>
> In Betty, we also provide several features, such as “gradient clipping” and “parameter callback”, that could be used to implement such inequality constraints in a simple (but potentially theoretically less strict) way. For example, when optimizing weight decay hyperparameters, whose values cannot be negative (i.e. $\lambda_{wd}\geq 0$), users can apply parameter clipping through the “parameter callback” method implemented within the **Problem** class. In addition to open-sourcing our library, we will release the extensive documentation for such features.
>
> > **Q7**  On p.3, when defining P1, should the last term of the argmin be D1 instead of Dk?
>
> **A7** Thanks for pointing out the typo! It should be $D_1$, and we modified our manuscript accordingly.
>
> > **Q8**  In B.2 (p. 15), both Correct and Reweight have exactly the same implementation for training_step, but the text and math in 5.2 (p.7-8) mention a different loss, Lval′, "augmented with the classification loss of the correction network", which I don't see in the code listing.
>
> **A8** In Appendix B, we attempt to maximally simplify our pseudo-code by omitting several details, and thereby allow readers to easily understand how the code written with Betty looks. However, it seems that such omission instead caused confusion, so we modified our manuscript to include them. Hope this clarified your concern!
>
> We again thank the reviewer for the thoughtful review. Please let us know if you have any further questions or comments!

---

> ### Author Response · Authors · 2022-11-12
> **Response to Reviewer MAHC 1/2**
>
> Thank you for the detailed review and positive feedback. In this response, we aim to address each of your comments.
>
> We would first like to highlight that we have updated our paper with two additional experiments, which further demonstrate the general applicability of Betty on large-scale datasets (e.g. Clothing-1M) and large models (e.g. BERT-base). These are described in Appendix A.2.
>
> ### **Clarity**
> > **Q1** It would be nice to have diagrams similar to Fig. 2 for the actual examples, especially 5.3 / B.3.
>
> **A1** Following the reviewer’s suggestion, we have included diagrams for each of our experiments in Appendix G. If the paper gets accepted, we will aim to move these diagrams to the main text given additional space for the camera-ready version.
>
> > **Q2** The direction of arrows / dependencies is not obvious to grasp, when they're presented as "dependencies". They may plausibly be "use -> definition" as well as "definition -> use".
>
> **A2** We agree that, from the new user’s perspective, it may not be very straightforward to determine the direction of arrows. We have made sure to emphasize/clarify the definition of these arrows in our paper (in paragraph left of Figure 2). We also find that if one writes down the mathematical formulation of the MLO program first, it helps in determining the arrow direction by seeing which level is constrained by (non-)optimal parameters from other levels.
>
> > **Q3** "training_step" is a confusing name when what seems to be described is a cost function, rather than an update step.
>
> **A3** While we agree that “training_step” may not be an ideal name for the cost function, we chose this name due to convention in the popular PyTorch-based software library (i.e. PyTorch Lightning), which uses this exact terminology for defining cost/loss function. Therefore, we hope that following this naming strategy will lower the barrier for new users who are already familiar with this related software package.
>
> > **Q4** How does it work when there is more than 1 top-level problem, as in 5.2 / B.2? Are both problems implicitly given a weight? It looks like there would usually be a trade-off.
>
> **A4** This is a great question and it’s totally up to the user's preference. By default, each top-level problem is given a uniform weight, though this could be weighted (by multiplying a factor to a problem’s cost function in the `training_step` method) to customize the MLO program for a given user or problem.
>
> > **Q5** It would probably help if indices i, j, k, l were used in a more consistent way (in which indices are upper vs. lower, for instance)
>
> **A5** Thanks for the feedback on this. Our goal was to use $i$ and $j$ for general indexing, $k$ for the problem level, and $l$ the lower level problem. To improve consistency, we have replaced $i$, $j$ in the paragraph left to Figure 2 on p4 to $P_i$ and $P_j$. If you have other suggestions that could improve the clarity of the paper, feel free to let us know!

---

### Official Review · Reviewer_gnFX · 2022-10-30

**Confidence:** 3
**Clarity, Quality, Novelty And Reproducibility:** 1. The paper is overall well-written …
**Correctness:** 4
**Technical Novelty And Significance:** 3
**Empirical Novelty And Significance:** 3
**Recommendation:** 8

**Strength And Weaknesses:**

Strength:
1. The efficient automatic differentiation method is novel and effective.
2. The proposed system BETTY is thoughtfully designed and clearly presented. It is a very meaningful contribution to facilitating efficient implementations of MLO programs. The authors also showcase the scalability of the proposed system to models with hundreds of millions of parameters by performing MLO on the BERT-base model.
2. The system will be released open source.
3. The background knowledge and math foundations are clearly presented.

This is very impressive work to me. I do not see a major limitation of the proposed system.

**Summary Of The Paper:**

This work presents an auto-differentiation library for gradient-based MLO. This work makes the following contributions:
1. This work develops an efficient automatic differentiation technique for LO based on a novel interpretation of MLO as a special type of dataflow graph. This technique can reduce the complexity of automatic differentiation from $O(d^3)$ to $O(d^2)$.
2. This work introduces a software library named BETTY to support large-scale MLO based on the technique mentioned in contribution 1.
The authors demonstrate the effectiveness and stability of BETTY with hundreds of millions of parameters by performing MLO on the BERT-base model.


**Summary Of The Review:**

This work proposes an automatic differentiation technique based on a novel interpretation of MLO as a special type of dataflow graph and then builds a software library for large-scale MLO based on the automatic differentiation technique. Both the automatic differentiation technique and the proposed system BETTY are very impressive contributions to facilitate further research and development on MLO. The paper is well-written and a pleasure to read.

---

> ### Author Response · Authors · 2022-11-12
> **Response to Reviewer gnFX**
>
> Thank you for the thoughtful review and positive feedback.
>
> We would like to highlight that we have updated our paper with two additional experiments, which further demonstrate the general applicability of Betty on large-scale datasets (e.g. Clothing-1M) and large models (e.g. BERT-base). These are described in Appendix A.2, and we copy the main results below:
>
> ### **Reweighting and Correcting Corrupted Labels w/ Clothing-1M + ResNet-50**
>
> |          | Test Accuracy |
> |----------|---------------|
> | Baseline | 70.76         |
> | +RWT     | 75.57         |
> | +RWT&CRT | **76.34**         |
>
>
> ### **Reweighting and Correcting Corrupted Labels w/ BERT-base + Wrench**
>
> |          | TREC        | AGNews      | IMDB        | SemEval     | ChemProt    | YouTube     |
> |----------|-------------|-------------|-------------|-------------|-------------|-------------|
> | Baseline | 64.14+-6.56 | 86.12+-0.17 | 71.66+-2.05 | 79.93+-1.53 | 52.35+-0.56 | 93.20+-1.44 |
> | +RWT     | 84.07+-4.42 | 89.62+-0.60 | **87.85**+-0.24 | 87.45+-0.69 | 71.42+-1.50 | 94.67+-0.46 |
> | +RWT&CRT | **93.07**+-0.31 | **90.40**+-0.16 | 87.45+-0.39 | **87.92**+-0.04 | **75.27**+-1.23 | **94.80**+-0.80 |
>
> We hope the above experiments make our work even stronger. If you have any other suggestions or comments for the paper, please do not hesitate to let us know!

---

### Decision · Program_Chairs · 2023-01-20

**Decision:**

Accept: notable-top-5%

**Justification For Why Not Higher Score:**

 N/A.

**Justification For Why Not Lower Score:**

 The oral recommendation for this paper is somewhat high for a "mostly" software-based paper, but in this case I think it's justified.  Both the technical developments and the showcasing of a usable and potentially very valuable software library makes this work likely to have a substantial impact.  I believe that there is quite a bit of value in highlighting these software efforts more to the broader ICLR community and thus (in addition to the well-received paper, as illustrated by the reviewer scores), I think this paper makes sense to present in this mode.

**Metareview: Summary, Strengths And Weaknesses:**

Thank you for your submission to ICLR.  This paper presents a library and associated methodological approaches for multilevel optimization.  The paper is quite valuable as both a methodological contribution (understanding dataflow graphs better for the multilevel optimization setting), as well as from a software engineering standpoint, where the paper evaluates in detail the design methodology.  All the reviewers were in agreement about the strength of this paper, and thus I recommend it for acceptance.

**Note From Pc:**

if the above contains the word "oral" or "spotlight" please see: "oral" presentation means -> notable-top-5% and "spotlight" means -> notable-top-25%. As stated in our emails, we are disassociating presentation type from AC recommendations

**Summary Of Ac-Reviewer Meeting:**

N/A